# Sex and gender considerations in implementation interventions to promote shared decision making: A secondary analysis of a Cochrane systematic review

Évèhouénou Lionel Adisso[1,2,3], Hervé Tchala Vignon Zomahoun[1,2,3,4], Amédé Gogovor[1,2,3,4], France Légaré[1,2,3,4]*

1 Canada Research Chair in Shared Decision Making and Knowledge Translation, Quebec, Canada,
2 Population Health and Practice-Changing Research Group, Université Laval Primary Care Research Centre (CERSSPL-UL), Quebec, Canada, 3 Department of Family Medicine and Emergency Medicine, Faculty of Medicine, Université Laval, Quebec, Canada, 4 Quebec SPOR SUPPORT UNIT, Quebec, Canada

* France.Legare@fmed.ulaval.ca

**Data Availability Statement:** All relevant data are within the manuscript and its Supporting Information files.

## Abstract

### Background

Shared decision making (SDM) in healthcare is an approach in which health professionals support patients in making decisions based on best evidence and their values and preferences. Considering sex and gender in SDM research is necessary to produce precisely-targeted interventions, improve evidence quality and redress health inequities. A first step is correct use of terms. We therefore assessed sex and gender terminology in SDM intervention studies.

### Materials and methods

We performed a secondary analysis of a Cochrane review of SDM interventions. We extracted study characteristics and their use of sex, gender or related terms (mention; number of categories). We assessed correct use of sex and gender terms using three criteria: "non-binary use", "use of appropriate categories" and "non-interchangeable use of sex and gender". We computed the proportion of studies that met all, any or no criteria, and explored associations between criteria met and study characteristics.

### Results

Of 87 included studies, 58 (66.7%) mentioned sex and/or gender. The most mentioned related terms were "female" (60.9%) and "male" (59.8%). Of the 58 studies, authors used sex and gender as binary variables respectively in 36 (62%) and in 34 (58.6%) studies. No study met the criterion "non-binary use". Authors used appropriate categories to describe sex and gender respectively in 28 (48.3%) and in 8 (13.8%) studies. Of the 83 (95.4%) studies in which sex and/or gender, and/or related terms were mentioned, authors used sex and

**Funding:** The author(s) received no specific funding for this work.

**Competing interests:** The authors have declared that no competing interests exist

gender non-interchangeably in 16 (19.3%). No study met all three criteria. Criteria met did not vary according to study characteristics (*p>.05*).

## Conclusions

In SDM implementation studies, sex and gender terms and concepts are in a state of confusion. Our results suggest the urgency of adopting a standardized use of sex and gender terms and concepts before these considerations can be properly integrated into implementation research.

## Introduction

Shared decision making (SDM) is an interpersonal, interdependent process in which health professionals and patients relate to and influence each other as they collaborate in making decisions about the patient's health [1–3]. Often supported by decision aids, SDM is based on best available evidence as well as the patient's values and preferences [4]. There is an ethical imperative to involve patients in making important health decisions [5] and SDM is appearing in legislation governing healthcare in numerous countries [6, 7]. SDM can improve patient engagement [8–11], satisfaction and adherence to drug therapy [12], and contributes to the optimization of health service utilization and health costs [13]. SDM improves patient experiences and the quality of care provided by health professionals [14]. Despite this potential, SDM is not implemented as much as it could be in clinical practice [15, 16]. A 2018 Cochrane systematic review on interventions to improve the use of SDM by health professionals highlighted that much remains to be done to identify more effective implementation interventions [16].

In recent years, implementation scientists have hypothesized that implementation interventions would be more effective if they incorporated considerations of sex and gender [17–19]. Sex and gender are important determinants of illness. A review exploring the role of sex and gender as modifiers of the most common causes of death and morbidity underlined many sex/gender-based differences [20]. According to authors, heart disease occurs in younger males with more obstructive coronary disease, whereas it occurs in older females with more coronary microvascular dysfunction. Furthermore, women are underdiagnosed for inflammatory airway disease, and have higher myocardial infarction mortality, fewer heart transplants (although they are more frequent donors) and overall receive less evidence-based treatment than men [20]. When findings for males and females are not disaggregated, results can hide important differences [21, 22]. Research has also shown differences in drug reactions and rehabilitation outcomes [23, 24]. Interventions that take sex and gender into consideration are thus likely to produce more reliable evidence. Indeed, if authors fail to consider potential differences in the effectiveness of an intervention for men and women, there is a risk of bias, since it has not accurately assessed for whom the intervention is effective [25]. Furthermore, the structural influence of sex and gender on other variables is often neglected. Yet a wide range of health variables are gendered, for example, occupational status, working conditions, and access to sexual health services [26, 27]. In implementation research, gender may be discussed under four constructs: gender roles, gender identity, gender relations and institutionalized gender. Each is associated with relevant measures, such as the Gender Role Conflict scale [28] and the Bem Sex Role Inventory [29]. Implementation studies that consider these constructs will thus improve outcomes such as acceptability, feasibility, adoption and sustainability [30].

Sex- and gender-considerations are important in both SDM itself and SDM implementation interventions. Sex and gender are important variables for decision making styles, communication styles, and values and preferences—all key issues in SDM [31–35]. Health professionals' sex and gender awareness will also impact their ability to identify risk factors for various illnesses, variables that may affect treatment options and their implications [36]. Failing to integrate sex and gender in SDM interventions such as training programs or decision aids neglects important determinants of knowledge use, reducing the effectiveness of the intervention, perhaps inadvertently reinforcing sex neutral claims and negative gender stereotypes, and thus perpetuating society's sex and gender inequities [17].

The first step in considering sex and gender in health research, including implementation research, is to ensure understanding of the terms and their appropriate use [21, 37]. The Canadian Institutes of Health Research (CIHR) and the U.S. National Institutes of Health (NIH) have proposed similar lexicons of appropriate sex and gender terms for health researchers. In their standardized terminologies, "sex" refers to a set of biological attributes in humans and animals. These attributes include physical and physiological features (including chromosomes), gene expression, hormone levels and function, and reproductive/sexual anatomy [38]. Sex is usually categorized as female or male, but there is variation in the biological attributes that comprise sex and how those attributes are expressed [39]. "Gender" refers to the socially constructed roles, behaviours, expressions and identities of girls, women, boys and men. Gender is usually conceptualized as a binary (woman/man or girl/boy), but there is considerable diversity in how individuals and groups understand, experience, and express gender: hence people can also be "gender diverse", e.g. transgender, agender, genderqueer or Two-Spirited [38]. Gender influences how people perceive themselves and each other, how they act and interact, and how power and resources are distributed in society [40]. The definitions may appear to categorize sex and gender as mutually exclusive, but they are interrelated and intersect [41]. While many interventions have been proposed to improve SDM among health professionals, little is known about how much they incorporate sex and gender. We aimed to take a first step by assessing variations in sex and gender concepts and terms in implementation studies of SDM in clinical practice.

## Materials and methods

### Study design

We performed a secondary analysis of the studies included in the qualitative synthesis of a Cochrane review on the effectiveness of interventions for increasing the use of SDM by health professionals [16]. There is no reporting guideline for secondary analyses of systematic reviews on the EQUATOR Network [42]. Thus we adapted the Preferred Reporting Items for Systematic Reviews and Meta-Analyses (PRISMA) for reporting our results [43].

### Search strategy and data sources

A detailed description of the search strategy, data sources can be found in the original Cochrane review [16].

### Criteria for including studies

We retained all 87 intervention studies included in the qualitative synthesis of the original review (Fig 1) [16]. These studies met the following criteria: a) **Study design** included randomized trials, non-randomized controlled trials, before-after studies, interrupted time series; b) **Participants** included health professionals in training or already trained who were responsible

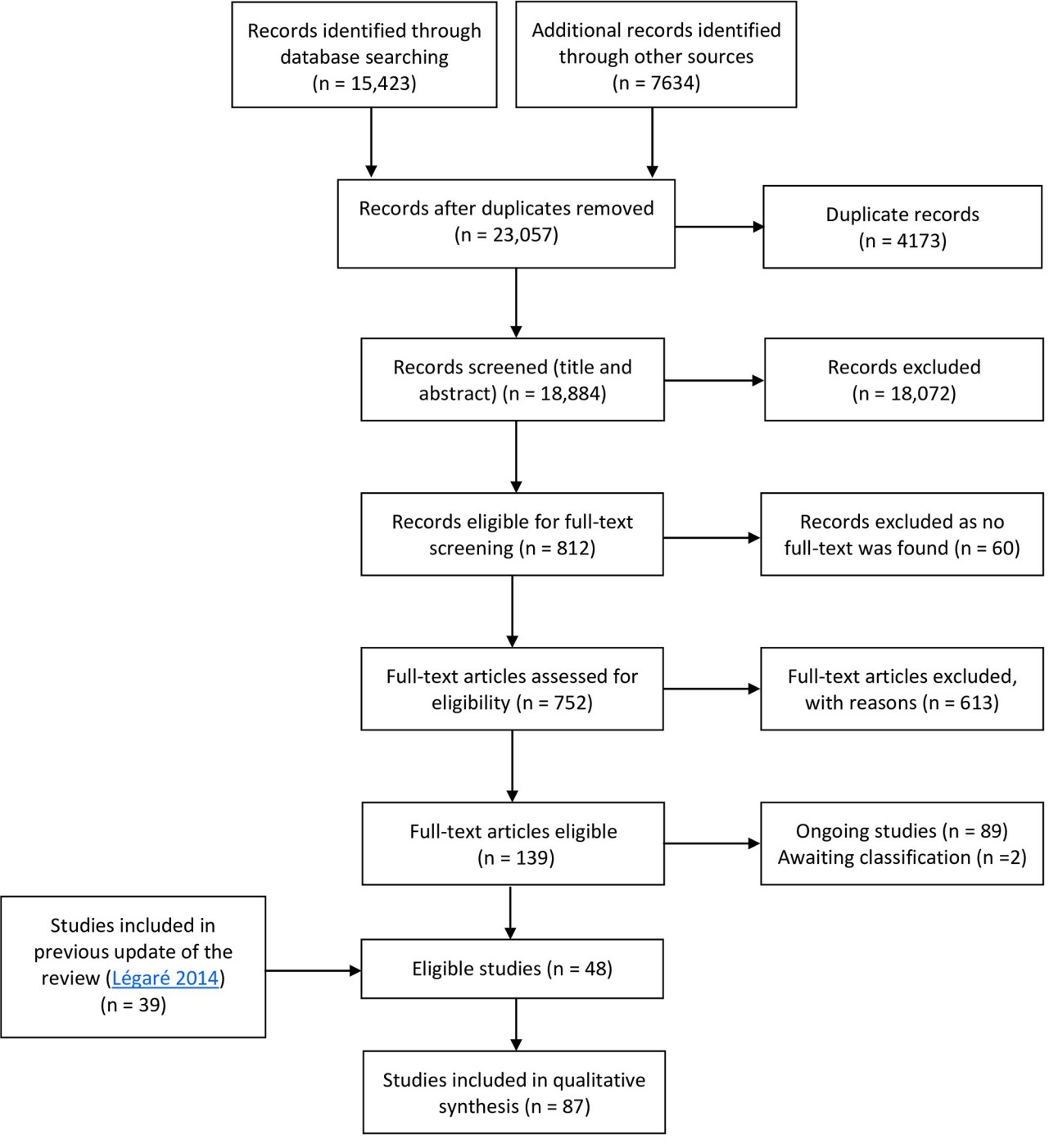

**Fig 1. Flow chart of the Cochrane systematic review [16].**

for the caring for patients, and patients of those same health professionals; c) **Intervention**s included any that was designed to promote the adoption of SDM by health professionals, such as training (e.g., conferences, workshops), distribution of printed educational materials (e.g., practice guidelines), audits with feedback, reminders, educational field visits, and patient-mediated interventions. The interventions targeted health professionals, patients or both;

d) **Comparisons** included usual care or other interventions targeting either health professionals, patients, or both; e) **Outcomes**, both patient-reported and observer-reported, concerned SDM that occurred during the meeting between patients (and their families if applicable) and the health professionals. These SDM outcomes could have been a primary or secondary outcome of the study.

## Process for assessing sex and gender considerations

To assess sex and gender considerations in each of the studies retained, we evaluated the correct use of such terms by measuring frequencies of the use of sex- and gender-related terms and determining categories used to further describe them. Then we assessed studies according to three criteria for correct use of sex and gender terms according to the definitions of sex and gender proposed by the CIHR [38] and the NIH [40]. These criteria were "non-binary use", "use of appropriate categories" and "non-interchangeable use of sex and gender".

**Non-binary use of sex and gender.** Sex can be male, female or intersex. While only two categories are usually used, intersex people are born with ambiguous reproductive or sexual anatomy. Two categories only are usually used for gender also, while gender is on a continuum and many people self-identify neither as men ("he") nor women ("she") but as pangender, transgender, gender-diverse etc. (and may prefer to be referred to as "they"). The criterion of "non-binary use" of sex or gender was therefore applied to assess whether studies took these important variations into consideration [44]. We assessed the criterion in studies in which the terms sex and/or gender (i.e. sex, gender or both) were mentioned. We considered the use of the terms as *binary* if authors mentioned the terms sex and/or gender and described them by using only two categories (i.e. male or female, man/boy or woman/girl). This included instances where categories were inferred, even if not explicitly mentioned (e.g., if authors only reported the proportion of female participants as 65%, we inferred that the proportion of male participants was 35% and considered that two categories were used). We considered the use as *non-binary* if authors mentioned the terms sex and/or gender and described them using a third category, i.e. 'intersex', 'gender diverse', or else provided a third sex or gender option by including the category 'not specified' or 'would prefer not to respond'. We considered the "non-binary use" as *unclear* if authors mentioned only the term sex or gender without specifying further categories (e.g. male, man, female, male, intersex, transgender).

**Use of the appropriate categories to describe sex and gender.** We assessed the "use of the appropriate categories" to describe sex and gender in studies in which sex and/or gender were mentioned. To assess this criterion, we considered the categories used to assess sex as *appropriate* if authors consistently used the categories male/female/intersex. Similarly, we considered the categories used to assess gender as *appropriate* if authors consistently used the categories "girl/woman", "boy/man", (i.e. identities that are culturally rather than biologically determined) and any word applying to gender-diverse people. We considered the categories used as *inappropriate* if authors mentioned the terms sex and/or gender but used the categories associated with sex to describe gender, and vice versa. We considered the categories used as *unclear* if authors mentioned only the term sex and/or gender without subdividing into categories.

**Non-interchangeable use of sex and gender.** To assess the "non-interchangeable use" of sex and gender, we referred to above-mentioned definitions of sex and gender as proposed by CIHR and NIH [38, 40]. We assessed this criterion in studies in which sex and/or gender and/or related terms were mentioned. We considered sex and gender were used *interchangeably* (non-interchangeable use = "No") if sex terms (sex, female, male, intersex) and gender terms (gender, girl, boy, woman, man, gender-diverse) were mentioned and indiscriminately used to

describe either sex or gender of participants in the same study. For example, authors may have used sex terms while describing the sociodemographic characteristics but reported or discussed the results using gender terms to describe the same attributes of the same participants. We considered sex and gender were used ***non-interchangeably*** (non-interchangeable use = "Yes") if sex terms were consistently used to describe biological attributes while gender terms were consistently used to describe sociocultural attributes of study participants. We considered the non-interchangeable use to be ***unclear*** (non-interchangeable use = "Unclear") in any other situation where the criteria were applicable.

   **Defining the "correct use" of sex and gender.**   The "correct use" of sex and gender was defined by combining all the three criteria: *non-binary use*, *use of appropriate categories* and *non-interchangeable use* of sex and gender. The "correct use" of sex and gender was then assessed in studies in which sex and/or gender were mentioned. We considered the use of sex and gender as ***correct*** if all three criteria were met. We considered the correct use of sex and gender as ***unclear*** if there was a combination of unclear and positive answers (e.g. *use of categories* = "appropriate" and *non-interchangeable use* = "unclear"). In any other situation, where the assessment was applicable, we considered the use as ***incorrect***.

## Data collection process

We screened the full text of each of the 87 studies included in the qualitative synthesis of the original Cochrane review [16]. Members of the research team discussed which data to extract. Data were extracted by a single reviewer (ELA) trained in sex and gender considerations in health research at Women's College Research Institute in Toronto, Ontario in March 2017 [45]. We first extracted data related to the characteristics of the included studies (e.g. author's name, year of publication, regions in which they were conducted, type of interventions, effect of the interventions on the primary outcome). Any ambiguities, unexpected or unanticipated issues faced during the data extraction were discussed with the other members of the research team. We then searched for terms related to the two key concepts: Sex and related terms such as "female", "male" and "intersex" [46, 47]; and gender and its related terms such as "women", "men", "woman", "man", "girl", "boy" and "gender diverse". We searched the full text either electronically or manually (when an electronic version of the paper was not available). We identified studies in which sex and/or gender and/or related terms were mentioned. We extracted data on the number and the wording of the categories used to describe sex and gender attributes and assessed the three criteria defined above (non-binary use, use of appropriate categories, non-interchangeable use of sex and gender).

## Data analysis

We performed a descriptive analysis for the characteristics of the 87 studies using frequency counts (number and percentage). The most recent changes in the definitions of sex and gender proposed by CIHR and NIH occurred around 2015 [38–40]. We therefore categorized "year of publication" as a binary variable: "before 2016" and "after 2016". We categorized the variable "regions in which the study was conducted" as a binary variable: "North America" and "Europe/Other". We described (using proportions) studies in which sex and/or gender and/or related terms were mentioned. We calculated the percentage of these studies that met each criteria of interest. We determined the proportion of studies with "correct use" of sex and/or gender, i.e. studies that met all three criteria. We compared the proportion of studies that met each criterion according to the following study characteristics: year of publication, regions in which the studies were conducted, type of intervention, and effect of intervention on primary outcome. To reach this goal, we reduced responses to each criterion from three to two:

"criteria was met" and "criteria was not met or unclear". We evaluated if this categorization would affect our results by performing sensitivity analyses with a third categorization (S1 Table). To explore associations between criteria met and study characteristics, we performed Pearson Chi-squared test [48], Yates's Khi-2 Correction for continuity [49] or Fischer's exact test [50]. All analyses were performed using version 9.4 of SAS software.

## Results

### Characteristics of included studies

Out of 87 included studies, all published between 1995 and 2017, almost a quarter (23.4%) were published in 2016 or 2017 [51–70]. The studies were mainly (54%) conducted in North America [52, 54–58, 61, 66–68, 71–105]. Forty-four (50.6%) studies evaluated interventions targeting patients [32, 51, 52, 54, 57, 59–64, 66, 67, 69, 72, 75–79, 84–86, 88, 90–92, 96, 98, 99, 102, 103, 106–116]. Fifteen (17.2%) studies evaluated interventions targeting health professionals [53, 55, 65, 94, 101, 109, 117–125]. Twenty-eight (32.2%) studies evaluated interventions in both patients and health professionals [56, 58, 68, 71, 73, 74, 83, 86, 89, 93, 95, 97, 99, 104, 105, 126–135]. Effect of the interventions (compared to usual care or the other types interventions by target group, i.e. patients, health professionals or targeting both) on the primary outcome was significant in 24 (28%) of the included studies [60, 64–66, 68–70, 87, 88, 92, 96, 101–103, 109, 113, 115, 117, 122, 124, 125, 129, 130, 136] (Table 1).

### Mention of sex and/or gender and/or related terms in included studies

Out of 87 included studies, the term, sex was mentioned in 37 (42.5%) studies [54–58, 60–64, 66, 68, 70, 71, 79, 83, 85, 87, 89, 95, 96, 98, 103, 105, 107, 110, 112, 114, 119–122, 126, 128, 130–132]. Gender was mentioned in 36 (41.4%) studies [51, 52, 58, 60, 61, 63, 65, 70, 71, 77, 82, 87, 89, 91, 96–99, 101, 104, 106, 112, 114, 117–119, 121, 124, 125, 131, 133–135, 137]. The terms sex and/or gender were mentioned in 58 (66.7%) studies [51, 53–66, 68, 70, 71, 77, 79, 82, 83, 85, 87–89, 91, 95–98, 101, 103–107, 110, 112, 114, 117–120, 122, 124–126, 128–135]. Terms related to sex and gender such as female, male, woman/women, man/men and girl were mentioned respectively in 53 (60.9%), 52 (59.8%), 38 (43.7%), 11 (12.6%) and 1 (1.2%) studies. The term boy was not mentioned in any study. The terms sex, gender and/or related terms were mentioned in almost all of the included studies: 83 (95.4%) [51, 52, 54–59, 64–68, 70–79, 82, 95–106, 109–118, 123–130, 134, 135, 137, 138]. Neither sex nor gender was mentioned in 29 (33.3%) studies [53, 67, 69, 72–76, 78, 81, 84, 86, 90, 92–94, 100, 102, 108, 109, 111, 113, 115, 116, 123, 127, 136–138]. Neither sex, gender nor any related term was mentioned in four (4.6%) studies [53, 69, 81, 136] (Table 1).

### Assessing the criteria for correct use of sex and gender

**Non-binary use of sex and gender.**   The non-binary use of sex and gender was assessed in the 58 studies in which sex and/or gender were mentioned [51, 53–66, 68, 70, 71, 77, 79, 82, 83, 85, 87–89, 91, 95–98, 101, 103–107, 110, 112, 114, 117–120, 122, 124–126, 128–135]. In these studies, authors clearly described sex as a binary variable in 36 (62.1%) studies [54–58, 60–64, 66, 68, 70, 71, 79, 83, 85, 87, 89, 95, 96, 98, 103, 107, 110, 112, 114, 118–122, 126, 130–132]. Such studies represented 97.3% of the 37 studies in which only sex was mentioned [54–58, 60–64, 66, 68, 70, 71, 79, 83, 85, 87, 89, 95, 96, 98, 103, 105, 107, 110, 112, 114, 119–122, 126, 128, 130–132]. The use of sex as a non-binary variable was unclear in 22 (37.9%) studies [51, 52, 59, 65, 77, 82, 88, 91, 97, 99, 101, 104–106, 117, 124, 125, 128, 129, 133–135] (Table 2). We found no studies whose use of sex was explicitly non-binary, i.e. we found no words or expressions to

Table 1. Characteristics of the 87 included studies and use of sex/gender terms.

| Characteristics | Number of studies | Percentage (%) |
|---|---|---|
| **Year of publication** | | |
| Before 2016[a] | 67 | 77.0 |
| After 2016[b] | 20 | 23.0 |
| **Regions in which the studies were conducted** | | |
| North America | 47 | 54.0 |
| Europe | 35 | 40.2 |
| Other (Australia and Namibia) | 5 | 5.8 |
| **Interventions targeting** | | |
| Patients | 44 | 50.6 |
| Health professionals | 15 | 17.2 |
| Both | 28 | 32.2 |
| **Effect of interventions on primary outcome** | | |
| Significant | 24 | 27.6 |
| Non-significant/Data not reported | 63 | 72.4 |
| **Mention of sex and/or gender and/or related terms** | | |
| **Sex and/or gender mentioned** | | |
| Sex | 37 | 42.5 |
| Gender | 36 | 41.4 |
| Sex and/or gender | 58 | 66.7 |
| Neither sex nor gender | 29 | 33.3 |
| **Related terms mentioned** | | |
| Female | 53 | 60.9 |
| Male | 52 | 59.8 |
| Woman/Women | 38 | 43.7 |
| Man/Men | 11 | 12.6 |
| Girl | 1 | 1.2 |
| **Sex and/or gender and/or related terms mentioned** | 83 | 95.4 |

[a] Before 2016: 1995–2015;

[b] After 2016: 2016–2017.

convey a non-binary conception of sex, such as "intersex", "not specified" or "would prefer not to respond" when authors were reporting selection of a sex option. Our findings were similar for gender. Out of the 58 studies in which sex and/or gender were mentioned, authors clearly used gender as a binary variable in 34 (58.6%) [39, 51, 52, 58, 59, 61, 63, 65, 70, 71, 77, 82, 88, 89, 91, 96–99, 101, 104, 106, 112, 114, 118, 119, 121, 124, 125, 129, 131–135]. This represented 94.4% of the 36 studies in which only gender was mentioned [51, 52, 58, 60, 61, 63, 65, 70, 71, 77, 82, 87, 89, 91, 96–99, 101, 104, 106, 112, 114, 117–119, 121, 124, 125, 131, 133–135, 137]. Non-binary use of gender was unclear in 24 (41.4%) studies [54–57, 60, 62, 64, 66, 68, 79, 83, 85, 87, 95, 103, 105, 107, 110, 117, 120, 122, 126, 128, 130]. We found no studies whose use of gender was explicitly non-binary, i.e we found no words or expressions to convey a non-binary conception of gender such as "gender-diverse" "transgender", "not specified" or "would prefer not to respond" when authors were reporting selection of gender options (Table 2).

**Use of the appropriate categories to describe sex and gender.** The appropriate categories used to describe sex and/or gender was assessed in the 58 studies in which sex and/or gender were mentioned. Out of these studies, authors used the appropriate categories (female/male) to describe sex in 28 (48.3%) studies [54–58, 61–64, 66, 68, 71, 83, 85, 87, 89, 95, 98, 103,

**Table 2. Assessing criteria of assessing a correct use of sex/gender.**

| Criteria | Number of studies | Percentage (%) |
|---|---|---|
| **1. Non-binary use of sex and gender (n = 58)** | | |
| **Sex** | | |
| Binary use | 36 | 62.1 |
| Non-binary use | 0 | 0 |
| Unclear use | 22 | 37.9 |
| **Gender** | | |
| Binary use | 34 | 58.6 |
| Non-binary use | 0 | 0 |
| Unclear use | 24 | 41.4 |
| **2. Use of appropriate categories (n = 58)** | | |
| **Sex** | | |
| Appropriate (Female/male) | 28 | 48.3 |
| Inappropriate (Woman/man) | 8 | 13.8 |
| Unclear | 22 | 37.9 |
| **Gender** | | |
| Appropriate (Woman/man) | 8 | 13.8 |
| Inappropriate (Female/male) | 26 | 44.8 |
| Unclear | 24 | 41.4 |
| **3. Non-Interchangeable use (n = 83)** | | |
| Yes | 16 | 19.3 |
| No | 48 | 57.8 |
| Unclear | 19 | 22.9 |
| **Correct use of sex and gender (n = 58)** | | |
| Correct use | 0 | 0 |
| Incorrect | 35 | 60.3 |
| Unclear | 23 | 39.7 |

107, 110, 112, 119, 121, 122, 130–132]. Such studies represented 77.8% of the 36 studies in which sex was used as a binary variable [54–58, 60–64, 66, 68, 70, 71, 79, 83, 85, 87, 89, 95, 96, 98, 103, 107, 110, 112, 114, 118–122, 126, 130–132]. Authors more often used the appropriate categories (female/male) to describe sex than the appropriate categories (woman/man) to describe gender. Out of the 58 studies in which sex and/or gender were mentioned, only 8 (13.8%) used the appropriate categories to describe gender [51, 70, 88, 91, 96, 114, 118, 133]. Such studies represented 23.5% of the 34 studies in which gender was mentioned [39, 51, 52, 58, 59, 61, 63, 65, 70, 71, 77, 82, 88, 89, 91, 96–99, 101, 104, 106, 112, 114, 118, 119, 121, 124, 125, 129, 131–135]. Authors mostly used the words female/male whether they were describing sex or gender (Table 2).

**Non-interchangeable use of sex and gender.** The non-interchangeable use of sex and gender was assessed in the 83 studies in which sex and/or gender and/or related terms were mentioned. Out of these studies, authors used sex and gender non-interchangeably in 16 (19.3%) studies [55, 62, 64, 66, 82, 85, 87, 89, 103, 107, 110, 119, 122, 126, 127, 130]. They used sex and gender interchangeably in 48 (57.8%) studies [51, 52, 54, 56, 58–61, 63, 65, 70, 71, 75, 77–79, 83, 84, 86, 88, 90, 91, 94–99, 101, 104–106, 109, 112, 114, 117, 118, 120, 121, 124, 125, 129, 131–135, 137], and unclearly in 19 (22.9%) studies [57, 67, 68, 72–74, 76, 92, 93, 100, 102, 108, 111, 113, 115, 116, 123, 128, 138] (Table 2).

**Correct use of sex and gender.** The correct use of sex and gender was assessed in studies where sex and/or gender were mentioned (n = 58) and in which all the criteria were applicable. None of these studies met all three criteria (Table 2), i.e. none of the included studies made the correct use of sex and gender. The use of sex and/or gender was incorrect in 35 (60.3%) [51, 52, 58, 59, 61, 63, 65, 70, 71, 77, 88, 89, 91, 96–99, 101, 104–106, 112, 114, 117–119, 121, 124, 125, 129, 131–135]. Their use was unclear in 23 (39.7%) studies [54–57, 60, 62, 64, 66, 68, 79, 82, 83, 85, 87, 95, 103, 107, 110, 120, 122, 126, 128, 130] (Table 2).

## Associations between criteria met and study characteristics

**Year of publication.** In studies published before 2016, the proportion that met the "use of appropriate categories" to describe "sex" was 41.5% (i.e. 17/41) [71, 83, 85, 87, 89, 95, 98, 103, 107, 110, 112, 119, 121, 122, 130–132] while it was 64.7% (i.e. 11/17) after 2016 [54–58, 61–64, 66, 68]. For the use appropriate categories to describe "gender" the proportion of studies that met this criterion was 14.6% [88, 91, 96, 114, 118, 133] before 2016 versus 11.8% [51, 70] after 2016. In studies published before 2016, the proportion that met the criterion: "non-interchangeable use of sex and gender" was 18.5% [82, 85, 87, 89, 103, 107, 110, 119, 122, 126, 127, 130] while it was 22.2% after 2016 [55, 62, 64, 66]. Year of publication made no significant difference to whether studies met the criteria of appropriate categories for sex ($p = .107$), gender ($p = .319$), or non-interchangeable use ($p = .984$) (Table 3).

**Regions in which studies were conducted.** Studies that met the "use of the appropriate categories" to describe "sex" in the ones conducted in North America was 53.3% (i.e. 16/30) [54–58, 61, 66, 68, 71, 83, 85, 87, 89, 95, 98, 103] versus 42.9% (i.e. 12/28) [62–64, 107, 110, 112, 119, 121, 122, 130–132] in Europe and other regions. For the appropriate categories used to describe "gender", the proportions of studies that met the criterion "use of the appropriate

**Table 3. Associations between criteria met and study characteristics.**

| Characteristics | Use of categories of sex (n = 58) | | | Use of categories of gender (n = 58) | | | Non-interchangeable use (n = 83) | | |
|---|---|---|---|---|---|---|---|---|---|
| | Appropriate | Inappropriate or Unclear | P-value | Appropriate | Inappropriate or Unclear | P-value | Yes | No or Unclear | P-value |
| **Year of publication** | | | .107[a] | | | .319[c] | | | .984[b] |
| <2016 | 17 (41.5) | 24 | | 6 (14.6) | 35 | | 12 (18.5) | 53 | |
| ≥2016 | 11 (64.7) | 6 | | 2 (11.8) | 15 | | 4 (22.2) | 14 | |
| **Regions in which studies were conducted** | | | .425[a] | | | .627[b] | | | .350[a] |
| North America | 16 (53.3) | 14 | | 3 (10.0) | 27 | | 7 (15.6) | 38 | |
| Europe/other | 12 (42.9) | 16 | | 5 (17.9) | 23 | | 9 (23.7) | 29 | |
| **Interventions targeting** | | | .619[a] | | | .737[a] | | | .771[a] |
| Patients | 13 (54.2) | 11 | | 5 (20.8) | 19 | | 7 (16.3) | 36 | |
| Health Professionals | 4 (36.4) | 7 | | 1 (9.1) | 10 | | 3 (23.1) | 10 | |
| Both | 11 (47.8) | 12 | | 2 (8.7) | 21 | | 6 (22.2) | 21 | |
| **Effect of interventions on primary outcome** | | | .670[a] | | | .672[c] | | | .427[b] |
| Significant | 7 (43.8) | 9 | | 3 (18.8) | 13 | | 6 (27.3) | 16 | |
| Non-significant/Data not reported | 21 (50.0) | 21 | | 5 (11.9) | 37 | | 10 (16.4) | 51 | |

[a] Pearson Chi-squared test;

[b] = Yates's Khi-2 Correction for continuity;

[c] = Fischer's exact test.

categories" to describe "gender" were respectively 10.0% [88, 91, 96] in North America versus 17.9% [51, 70, 114, 118, 133] in Europe and other regions. The proportion of studies conducted in North America in which authors used sex and gender non-interchangeably was 15.6% [55, 66, 82, 85, 87, 89, 103] versus 23.7% [62, 64, 107, 110, 119, 122, 126, 127, 130] in Europe and other regions. Regions in which studies were conducted made no significant difference to whether studies met the criteria of "use of the appropriate categories for sex ($p = .425$), gender ($p = .627$) or "non-interchangeable use of sex and gender" ($p = .350$) (Table 3).

**Type of interventions.**    In studies with significant interventions effect, the proportion in which authors used the appropriate categories to describe "sex" were respectively 54.2% (i.e. 13/24) [54, 57, 61–64, 85, 98, 103, 107, 110, 112] for interventions targeting patients, 36.4% (i.e. 4/11) [55, 119, 121, 122], targeting health professionals and 47.8% (i.e. 11/23) [56, 58, 68, 71, 87, 89, 95, 130–132] targeting both. For the appropriate categories to describe "gender", the proportions were respectively 20.8% [51, 88, 91, 96, 114], 9.1% [118] and 8.7% [70, 133]. Seven (16.3%) [62, 64, 66, 85, 103, 107, 110] of studies that met the criterion "non-interchangeable use of sex and gender" evaluated interventions targeting patients, 3 (23.1%) [55, 119, 122] targeting health professionals and 6 (22.2%) targeting both [82, 87, 89, 126, 127, 130]. The type of interventions made no significant difference to whether studies met the criteria of "use of the appropriate categories" for sex ($p = .619$), gender ($p = .737$), or non-interchangeable use ($p = .771$) (Table 3).

**Efficacy of interventions compared to usual care/other interventions on the primary outcome.**    The proportion of studies with significant interventions effect in which authors used the appropriate categories to describe "sex" was 43.8% (i.e. 7/16) [64, 66, 68, 87, 103, 122, 130] versus 50.0% (i.e. 21/42) [54–58, 61–63, 71, 83, 85, 89, 95, 98, 107, 110, 112, 119, 121, 131, 132] with non-significant interventions effect. For the appropriate categories to describe "gender", the proportions were respectively 18.8% [70, 88, 96] versus 11.9% [51, 91, 114, 118, 133]. The proportion of studies that evaluated effective interventions in which authors used sex and gender non-interchangeably was 27.3% [64, 66, 87, 103, 122, 130] versus 16.4% [55, 62, 82, 85, 89, 107, 110, 119, 126, 127] of studies that evaluated non-effective interventions in which authors used sex and gender non-interchangeably. The efficacy of interventions made no significant difference to whether studies met the criteria of appropriate categories for sex ($p = .670$), appropriate categories for gender ($p = .672$), or non-interchangeable use ($p = .427$) (Table 3).

## Discussion

This study assessed the use of sex and gender terms in 87 implementation intervention studies promoting adoption of SDM in clinical practice. Most authors made some mention of the terms sex and/or gender and/or related terms to describe study participants. The related terms they mostly mentioned were female and male. No authors used sex or gender as non-binary variables. More studies used appropriate categories to describe sex than to describe gender. Sex and gender were used synonymously or interchangeably in most studies. No single study met all criteria for "correct use" of the terms for reporting on sex and gender, i.e. use that is non-interchangeable, appropriately categorized, and non-binary. The proportion of studies meeting the criteria did not vary significantly according to publication year, region, intervention type or efficacy of interventions per target population. These results lead us to make the following observations.

First, authors did not use the terms "sex" and "gender" systematically when describing sociodemographic characteristics of study participants. Many switched back and forth, using them interchangeably, i.e. synonymously. These authors are neglecting an important

distinction in the implementation of SDM interventions and create confusion about to whom these interventions are applicable. [41, 139, 140]. Gender was frequently used as a synonym for sex and, even in single-sex studies, investigators consistently considered female (or male) systematically as women (or men) without assessing gender [140]. The studies that do not fall into this trap cannot be associated with a particular time, place, intervention type, or even the efficacy of their interventions: in our study the non-interchangeable use of sex and gender did not vary according to these study characteristics. Second, the variables "sex" and "gender" were not reported according to definitions used by NIH and CIHR [38–40]. The necessity for researchers applying to CIHR (after 2010) and NIH (after 2015) grants to solidly and accurately integrate sex and gender in their research proposals [141] has clearly not yet had an impact on implementation scientists' efforts to increase the adoption of SDM by health professionals [142]. These requirements in grant applications, training on sex and gender in research, and involving graduate students in sex and gender networks will presumably improve understanding and appropriate use of sex and gender in time [143, 144].

Third, all studies used sex and gender as binary variables. Statistics Canada, a common reference for country-wide data on health and a multitude of other variables, adopted the use of sex and gender as non-binary variables in 2018 [46, 145]. Non-binary people are an important and increasingly vocal segment of the population with their own specific physical and mental health issues. Many already feel excluded by the health system, and being invisible in research results will be a further alienating factor [17].

Regarding measurement, authors in our studies did not report on how the sex of participants was measured. Most of the evaluated interventions occurred in clinical care consultations, and some data on study participants' sex could be have been accessed using medical charts. If investigators found no known intersex study participants (intersex individuals are fewer than 1 in 2000) [47], they may have probed no further and seen little point reporting a category with zero individuals. Measuring the gender of participants is more challenging, yet equally important. One recent study on cardiovascular risk factors measured gender using information about whether the respondent was the primary household earner, their income, number of housework hours, and stress levels at home, as well as measures of masculinity and femininity from the Bem Sex-Role Inventory [29]. It found that both sex and gender were important in predicting many cardiovascular risk factors, but that the gender score was generally more important [146]. While it has been suggested that gender be operationalized through the four constructs of gender roles, gender identity, gender relations and institutionalized gender [139, 147, 148], it is still not clear how best to capture them. It is not clear if gender is a categorical variable, as Statistics Canada suggested in 2018, describing the variable "gender" with the categories "man", "woman" and "gender diverse" [145], or if gender should be assessed by scoring through a scale (continuous variable), as suggested by some previous works [29, 149–152]. Measuring gender in secondary data analyses such as ours, where direct measures of gender have not been collected, is even more challenging than in primary studies, as there is no access to the questionnaires used by investigators to understand how the variables were collected and categories defined. Smith et al. used the Labour Force Survey to develop their gender index, because it has questions that are commonly available in other data sources, and a gender-index using such data may be easily applied (and modified or further developed) to other secondary data sources [153]. With Statistics Canada now using non-binary categories, it could also be an interesting source of data for developing a more representative gender-index for further secondary studies in SDM.

Sex and gender are interrelated and dividing them up is misleading. With Tannenbaum et al. [17], we agree that SDM intervention studies should ask a question about "sex assigned at birth" and follow up with a gender question about "how the participant identifies him/herself

now" [17]. This will help the participant first to be aware of the fact that it is two different measures, and second to feel that the answer can be diverse. We suggest providing categories to describe both sex ("female", "male", "intersex", "do not want to respond", "other (please explain)" and gender ("girl/woman", "boy/man", "gender diverse", "do not want to answer", "other (please explain)".

To the best of our knowledge, this study may be one of the first studies to evaluate correct use of sex and gender by identifying relevant criteria. These criteria could be transformed into questions that authors can answer to in order to assess whether the first step in sex and gender considerations has been understood.

## Conclusions

SDM interventions that do not consider sex and gender miss important biological and cultural differences between people that have a significant impact on health and health communications about health. The first step in correcting this is a good understanding and an appropriate use of sex and gender terms. We established criteria on correct use of terms and found that few studies of interventions to improve SDM in health professionals correctly identified sex and gender and their categories, and none described them as non-binary. Standardizing terminology would be a good start for measuring and reporting on sex and gender in SDM implementation interventions.

## Supporting information

**S1 Checklist. PRISMA 2009 checklist.**
(DOC)

**S1 Table. Description of the criteria according to the studies characteristics (sensitivity analyses).**
(DOCX)

**S1 Database.**
(XLSX)

## Acknowledgments

We thank Maude Downey and Louisa Blair for her editorial support.

## Author Contributions

**Conceptualization:** Évèhouénou Lionel Adisso, France Légaré.

**Data curation:** Évèhouénou Lionel Adisso, Hervé Tchala Vignon Zomahoun.

**Formal analysis:** Évèhouénou Lionel Adisso.

**Investigation:** Évèhouénou Lionel Adisso, France Légaré.

**Methodology:** Évèhouénou Lionel Adisso, Hervé Tchala Vignon Zomahoun, Amédé Gogovor, France Légaré.

**Resources:** Évèhouénou Lionel Adisso, France Légaré.

**Supervision:** Hervé Tchala Vignon Zomahoun, Amédé Gogovor.

**Validation:** Hervé Tchala Vignon Zomahoun, France Légaré.

**Writing – original draft:** Évèhouénou Lionel Adisso.

**Writing – review & editing:** Évèhouénou Lionel Adisso, Hervé Tchala Vignon Zomahoun, Amédé Gogovor, France Légaré.

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
