## [Decision Letter · Decision Letter 0]

29 Jul 2020

PONE-D-20-16595

Sex and gender considerations in implementation interventions to promote shared decision making: A secondary analysis of a cochrane systematic review

PLOS ONE

Dear Dr. Légaré,

Thank you for submitting your manuscript to PLOS ONE. After careful consideration, we feel that it has merit but does not fully meet PLOS ONE’s publication criteria as it currently stands. Therefore, we invite you to submit a revised version of the manuscript that addresses the points raised during the review process.

This is a well conceived piece which has been relatively well executed and reported.

The Reviewer has made some very sensible and constructive points which will enhance the value of your  paper to a more general audience. Given the importance of the subject matter being understood widely, I very much hope you can work with this.

We look forward to receiving your revised manuscript.

Kind regards,

Robert S. Phillips

Academic Editor

PLOS ONE

Journal Requirements:

Reviewers' comments:

Reviewer's Responses to Questions

**Comments to the Author**

1. Is the manuscript technically sound, and do the data support the conclusions?

Reviewer #1: Yes

2. Has the statistical analysis been performed appropriately and rigorously? 

Reviewer #1: I Don't Know

3. Have the authors made all data underlying the findings in their manuscript fully available?

Reviewer #1: Yes

4. Is the manuscript presented in an intelligible fashion and written in standard English?

Reviewer #1: Yes

5. Review Comments to the Author

Reviewer #1: PLOS ONE Review: Sex and gender considerations in implementation interventions to promote shared decision making: a secondary analysis of a Cochrane systematic review

Overall

• The authors review the extent to which studies on shared decision making reported on sex and gender of participants, and whether that reporting was appropriate. They make the point that sex and gender matter for the effectiveness of SDM interventions, and thus it is important to understand the extent to which research on SDM interventions has accounted for sex and gender.

• Throughout, the authors should make much more clear the evidence for WHY taking into account sex and gender in shared decision-making matters (of course, I believe that it does matter), but the case needs to be made more clearly with specific examples as to why it is essential to accurately measure, report, and discuss these constructs as the distinct characteristics that they are (sex vs gender identity). The whole value of the paper rests on people understanding and accepting WHY sex and gender considerations matter for SDM work.

• These results are of interest and make a contribution, however, in their current form they are too vague and unclear as to make that contribution effectively. Substantial revisions to the writing, to the level of detail provided, and clearer examples and arguments from the authors would go a long way toward making the paper a substantive contribution to the SDM field.

• The language/style of writing throughout the paper is not always clear, and in some cases is grammatically incorrect. The paper would benefit from a thorough review for language flow/accuracy.

• Inconsistent capitalization of sex/gender throughout paper – be consistent.

Abstract

• In background section of the abstract, the authors need to provide their rationale for WHY sex and gender matter for implementation interventions. Also, SDM should be defined and contextualized – are you looking at SDM in all fields? Only certain fields? Certain subpops? All patient populations?

• Materials and methods: While challenging in a short space, more clarity about what was measured and how assessed is needed. Language is vague.

• Results: “sex/gender” in line 43 – is this that sex OR gender was measured? Or both? This is an important distinction that should be clarified.

Introduction

• The introduction should be broken up into several paragraphs to improve readability/flow. For instance, the first paragraph could focus first on defining SDM, the second on WHY SDM is important in clinical care, the third on research/ideas related to how gender and sex are essential to intervention effectiveness, and so on (rather than having all of the above in the first paragraph as currently stands)

• Line 64-65: What is meant by “SDM can also facilitate care…” – what is meant by facilitating care? Give examples. Also – if I interpret correctly, this sentence may be somewhat redundant to the prior sentence – if they are addressing different things, be specific – what does this sentence add?

• Lines 75-76: The authors write that sex and gender can offer insight into how to tailor implementation interventions for greater efficacy – this is ESSENTIAL to the purpose of the paper, so the authors would do well to add several sentences diving in to this. HOW can sex and gender help increase intervention effectiveness? What specific studies have found this? How did they measure sex/gender? What did they find? What is the causal mechanism involved with this hypothesized pathway?

• In lines 80-99, it would help to put quotes around the terms you are defining. This paragraph is hard to follow.

• For “gender diverse”, give examples of gender identities for the unfamiliar research (i.e., agender, nonbinary, transgender, Two-spirit, genderfluid, pangender, etc)

• Authors periodically begin sentences with “It” and seem to be referring to sex or gender, however it is hard to follow – be more specific.

• Line 96-97: “All individuals act in ways that fulfill the gender expectations…” – definitely not true. Many people act in ways that DO NOT fulfill the gender expectations of their society – this is part of why critical gender analysis is such a focus. What is meant by this original sentence? Context is lacking.

Materials and methods

• Criteria for including studies: lines 134-135: section on comparison with another intervention being absent – this does not make sense. Does this mean only studies that did NOT compare the intervention to another intervention were included? Why is that? What is the rationale? Or if it does not mean this, what does it mean?

• Language unclear throughout. More detail/specifics needed.

• Process for assessing sex and gender considerations: more clarity needed throughout. Where possible, give specific examples. For example – the statement in line 143: “First, sex and gender are non-binary variables.” This section is setting up for later sections with more detail, but on first read, it leaves the reader wanting a lot more information right from the start. Either clarify that this information will be provided later, or consider streamlining this section and just diving directly into the more specific sub-sections that follow.

• Lines 147-148: be careful with language here. What is mean by “typical definition” of female or male?

Data collection process

• Lines 227-230: What software (if any) was used to manage and analyze extracted data?

Data analysis

• Line 243: list explicitly the characteristics meant by “characteristics of studies” to summarize/make obvious for reader: i.e., “….according to the characteritics of studies, which included: year of publication, region in which study conducted, study population, etc…”

• Authors list the statistical tests performed, but do not indicate which tests were used for which assessments. More detail is needed to evaluate the appropriateness of testing conducted.

Results

• Line 269: “sex/gender” again used, without clarity as to whether this mean one or the other, versus BOTH

• Table 1 and Table 3:

o for year of publication, give range of years (i.e., 2000-2015, vs 2016-2017) for specificity

o Instead of saying “countries”, say “regions” as you are comparing on region, not country

o Order patients/hc professionals above “both”, as otherwise “both” is confusing before you know what the other options are

o “sex and/or gender” is confusing in this table – shouldn’t it be, “sex AND gender”?

o Why is “boy” not included in Table 1 under “related terms mentioned”

• Language unclear/vague/incorrect in some sentences: i.e., Line 293-294 “Authors used sex as non-binary variable in no study”. Same for lines 307-308, 331-332, 373, etc

• Table 2:

o What is “respect of the criteria (2 and 3)” representing? What does this mean?

• Line 354: “Countries” should be “regions” throughout

Discussion

• Overall, the discussion should be shortened/tightened, remove speculation, and provide MUCH more specific commentary. A thorough grammar review is also warranted, especially on page 24

• More is needed to discuss the implications of sex and gender (measurement and reporting) for shared decision-making. Why does sex/gender matter for SDM? What has this review taught us about state of sex/gender measurement/reporting in SDM intervention studies? What are the implications of these findings for future SDM work? The authors mention these things but very vaguely – instead, the authors should add detail to provide information on specific findings from studies that have established the importance of sex/gender for SDM, and specific suggestions for how future SDM research should measure/report on sex/gender as a result of these findings, and why

• Feels that discussion could be broadly restructured/focused as follows:

o First: succinct narrative summary of main findings: no study met all three criteria for correct reporting on sex/gender. This did not vary by measured study characteristics.

o Second: implications of this finding for existing SDM research, and future SDM research

o Third: strengths/limitations

o Fourth: conclusions � recommendations from authors based on findings

• More specifically: authors could recommend measures of sex and gender that SDM researchers could use going forward – there is a rich literature on measurement of sex and gender identity that they could draw from and cite

6. PLOS authors have the option to publish the peer review history of their article (what does this mean?). If published, this will include your full peer review and any attached files.

Reviewer #1: No

---

## [Author Response · Author response to Decision Letter 0]

22 Sep 2020

Reviewer #1: PLOS ONE Review: Sex and gender considerations in implementation interventions to promote shared decision making: a secondary analysis of a Cochrane systematic review

Overall

• The authors review the extent to which studies on shared decision making reported on sex and gender of participants, and whether that reporting was appropriate. They make the point that sex and gender matter for the effectiveness of SDM interventions, and thus it is important to understand the extent to which research on SDM interventions has accounted for sex and gender.

• Throughout, the authors should make much more clear the evidence for WHY taking into account sex and gender in shared decision-making matters (of course, I believe that it does matter), but the case needs to be made more clearly with specific examples as to why it is essential to accurately measure, report, and discuss these constructs as the distinct characteristics that they are (sex vs gender identity). The whole value of the paper rests on people understanding and accepting WHY sex and gender considerations matter for SDM work. 

Authors' response:

Thank you for your comments. We addressed your comments, had an English-language editor revise the whole paper. We particularly revised the Introduction and Discussion to put more emphasis on why taking sex and gender in SDM interventions matter and on the implications of our findings (Pages 2-39, lines 29-814).

Comment: 

• These results are of interest and make a contribution, however, in their current form they are too vague and unclear as to make that contribution effectively. Substantial revisions to the writing, to the level of detail provided, and clearer examples and arguments from the authors would go a long way toward making the paper a substantive contribution to the SDM field.

Authors' response:

We revised the writing, increased the level of detail and provided clearer examples (Pages 17-29, lines 183 - 597).

Comment: 

• The language/style of writing throughout the paper is not always clear, and in some cases is grammatically incorrect. The paper would benefit from a thorough review for language flow/accuracy. 

Authors's response:

An English-language editor has revised the paper (Pages 2-39, lines 28-475).

Reviewer's comment:

• Inconsistent capitalization of sex/gender throughout paper.

Authors' response:

We corrected this throughout (Pages 2-39).

Reviewer's comment:

Abstract

• In background section of the abstract, the authors need to provide their rationale for WHY sex and gender matter for implementation interventions. Also, SDM should be defined and contextualized – are you looking at SDM in all fields? Only certain fields? Certain subpops? All patient populations? 

Authors' response:

Thank you for your comment. We defined SDM in healthcare and explained why sex and gender matter as follows:

Shared decision making (SDM) in healthcare is an approach in which health professionals support patients in making decisions based on best evidence and their values and preferences. Considering sex and gender in SDM research is necessary to produce precisely-targeted interventions, improve evidence quality and redress health inequities. We provided more details in the background.

There were no restrictions on the patient populations. (Page 2, lines 30-40).

Revierwer's comment:

• Materials and methods: While challenging in a short space, more clarity about what was measured and how assessed is needed. Language is vague. 

more clarity about what was measured and how assessed is needed.

Authors' response:

Thank you for comment. We provided more clarification about what was measured and assessed as follows:

We performed a secondary analysis of a Cochrane review of SDM interventions. We extracted study characteristics and their use of sex, gender or related terms (mention; number of categories). We assessed correct use of sex and gender terms using three criteria: “non-binary use”, “use of appropriate categories” and “non-interchangeable use of sex and gender”. We computed the proportion of studies that met all, any or no criteria, and explored associations between criteria met and study characteristics (Page 2-3, lines 41-52).

Reviewer's comment: 

• Results: “sex/gender” in line 43 – is this that sex OR gender was measured? Or both? This is an important distinction that should be clarified.

Authors'response:

Thank you for pointing this out. We corrected this as follows: sex and/or gender; and applied throughout the paper and tables (Page 3, lines 53-64).

Revierwer's comment:

Introduction

• The introduction should be broken up into several paragraphs to improve readability/flow. 

Separate into many other paragraphs to make it easy to read.

For instance, the first paragraph could focus first on defining SDM, the second on WHY SDM is important in clinical care, the third on research/ideas related to how gender and sex are essential to intervention effectiveness, and so on (rather than having all of the above in the first paragraph as currently stands)

Authors's response:

Thank you for your comment. We have extensively rewritten the Introduction. As suggested, we separated it into four paragraphs and four themes: The first paragraph is now about SDM (definition and importance clinical practice). 

The second is about why sex and gender matter in health research, implementation research and specifically in SDM and SDM interventions.

The third is about sex and gender concepts and definitions according to the CIHR and the NIH, and underlines that the first step in sex and gender considerations is defining and using the terms appropriately (Pages 4-9, lines 74-198).

Reviewer's comment:

• Line 64-65: What is meant by “SDM can also facilitate care…” – what is meant by facilitating care? Give examples.

Also – if I interpret correctly, this sentence may be somewhat redundant to the prior sentence – if they are addressing different things, be specific – what does this sentence add? 

Authors' response:

Thank you for your comment. We removed this sentence.

Rewiewer' comment:

• Lines 75-76: The authors write that sex and gender can offer insight into how to tailor implementation interventions for greater efficacy – this is ESSENTIAL to the purpose of the paper, so the authors would do well to add several sentences diving into this. HOW can sex and gender help increase intervention effectiveness? What specific studies have found this? How did they measure sex/gender? What did they find? What is the causal mechanism involved with this hypothesized pathway? 

Author's response:

Thank you for your comment. We addressed the complex question of sex and gender in implementation research in more depth in the 2nd paragraph, as follows:

“In recent years, implementation scientists have hypothesized that implementation interventions would be more effective if they incorporated considerations of sex and gender (Tannenbaum et al., 2016; Oertelt-Prigione et al., 2011). Sex and gender are important determinants of illness. A review exploring the role of sex and gender as modifiers of the most common causes of death and morbidity underlined many sex/gender-based differences (Mauvais-Jarvis et al., 2020). According to authors, heart disease occurs in younger males with more obstructive coronary disease, whereas it occurs in older females with more coronary microvascular dysfunction. Furthermore, women are underdiagnosed for inflammatory airway disease, and have higher myocardial infarction mortality, fewer heart transplants (although they are more frequent donors) and overall receive less evidence-based treatment than men (Mauvais-Jarvis et al., 2020) Thus when findings for males and females are not disaggregated, results can hide important differences (Day et al., 2016; De Castro et al., 2016) Research has also shown differences in drug reactions and rehabilitation outcomes (Robles et al., 2014; Tamargo et al., 2017) . Interventions that take sex and gender into consideration are thus likely to offer more reliable evidence. Moreover, if authors fail to consider potential differences in the effectiveness of an intervention for men and women, there is a risk of bias, since it has not accurately assessed for whom the intervention is effective (Runnels et al., 2014). Furthermore, the structural influence of sex and gender on other variables is often neglected. Yet a wide range of health variables are gendered, for example, occupational status, working conditions, and access to sexual health services. (Campos-Serna, Ronda-Pérez et al. 2013)

Pelletier R et al. 2015) In implementation research, gender may be discussed under four constructs: gender roles, gender identity, gender relations and institutionalized gender. Each is associated with relevant measures, such as the Gender Role Conflict Index (O'Neil et al., 2013) and the Bem Sex Role Inventory (Bem et al., 1977).Implementation studies that consider these constructs will thus improve outcomes such as acceptability, feasibility, adoption and sustainability (Peters et al., 2013) (Pages 4-6, lines 92-121).

Reviewer's comment:

• In lines 80-99, it would help to put quotes around the terms you are defining. This paragraph is hard to follow. 

Authors's response: 

Corrected as suggested (Page 7-8, lines 159-179).

Reviewer's comment:

• For “gender diverse”, give examples of gender identities for the unfamiliar research (i.e., agender, nonbinary, transgender, Two-spirit, genderfluid, pangender, etc.)

Authors' responses:

Authots' comment:We gave examples as suggested (Page 8, lines 172-173).

Reviewer' comment:

• Authors periodically begin sentences with “It” and seem to be referring to sex or gender, however it is hard to follow – be more specific. 

Authors' response:

We corrected this and added more concrete details (Pages 4-9, lines 74-198).

Reviewer's comment:

• Line 96-97: “All individuals act in ways that fulfill the gender expectations…” – definitely not true. Many people act in ways that DO NOT fulfill the gender expectations of their society – this is part of why critical gender analysis is such a focus. What is meant by this original sentence? Context is lacking . 

Authors' response: 

Thank you for pointing this out. We were trying to express societal expectations of normative behaviour but worded it wrongly. We have removed the phrase.

Reviewer's comment:

Materials and methods

• Criteria for including studies: lines 134-135: section on comparison with another intervention being absent – this does not make sense. Does this mean only studies that did NOT compare the intervention to another intervention were included? Why is that? What is the rationale? Or if it does not mean this, what does it mean?

• Language unclear throughout. More detail/specifics needed.

Authors' responses:

Thank you for your comment. We corrected this as follows: 

Comparisons included usual care or other interventions targeting either health professionals, patients or both.

We clarified our language and provide more details

Reviewer's comment:

• Process for assessing sex and gender considerations: more clarity needed throughout. Where possible, give specific examples. For example – the statement in line 143: “First, sex and gender are non-binary variables.” This section is setting up for later sections with more detail, but on first read, it leaves the reader wanting a lot more information right from the start. Either clarify that this information will be provided later or consider streamlining this section and just diving directly into the more specific sub-sections that follow

Authors'response:

Thank you for your comment. We have substantially rewritten this section. We reduced the introductory “Process” paragraph as follows:

To assess sex and gender considerations in each of the studies retained, we evaluated the use of such terms by measuring frequencies of the use of sex- and gender-related terms and determining categories used to further describe them. Then we assessed studies according to three criteria for the appropriate use of sex and gender terms according to the definitions of sex and gender proposed by the CIHR (27) and the NIH (29). These criteria were “non-binary use”, “use of appropriate categories” and “non-interchangeable use of sex/gender".

We moved the rationale for each criterion into the specific subsections that follow (Page 11, lines 230-238).

Reviewer's comment: 

• Lines 147-148: be careful with language here. What is mean by “typical definition” of female or male? Data collection process.

Authors' response:

We removed this phrase. 

Reviewer comment:

• Lines 227-230: What software (if any) was used to manage and analyze extracted data? 

Authors' response: 

We provided this in the section “data analysis”. We used SAS Software to analyze extracted data (Page 17, line 381).

Reviewer's comment:

Data analysis

• Line 243: list explicitly the characteristics meant by “characteristics of studies” to summarize/make obvious for reader: i.e., “….according to the characteristics of studies, which included: year of publication, region in which study conducted, study population, etc…” 

Authors' response:

Thank you for your comment. We specified the characteristics (Page17, lines 370-372).

Reviewer' comment:

• Authors list the statistical tests performed, but do not indicate which tests were used for which assessments. More detail is needed to evaluate the appropriateness of testing conducted.

Author's response: 

Thank you for your comment. We indicated the tests and the assessment for which we used them in data analysis section and in a footnote of Table 3. We also provided references about each test (Page 17, lines 478-380;

Table 3 : Page 29, line 597).

Reviewer's comment:

• Line 269: “sex/gender” again used, without clarity as to whether this mean one or the other, versus BOTH.

Authors's response: 

Thank you for your comment. We have corrected this to sex and/or gender meaning sex, gender or both. We defined it at its first use in the text and used this term throughout (Page 12, Line 272).

Reviewer's comment:

• Table 1 and Table 3:

o for year of publication, give range of years (i.e., 2000-2015, vs 2016-2017) for specificity 

Authors's response: 

Thank you for the precision. We provided the range of the year of publication in a footnote (Pages 19 and 29).

Reviewer's comment:

o Instead of saying “countries”, say “regions” as you are comparing on region, not country

Authors's response:

We replaced “countries” with “regions” (Pages 19 and 29).

Reviewer's comment:

o Order patients/hc professionals above “both”, as otherwise “both” is confusing before you know what the other options are 

Authors's response:

Thank you for your comments. We ordered categories as suggested (Pages 19 and 29).

Reviewer's comment:

o “sex and/or gender” is confusing in this table – shouldn’t it be, “sex AND gender”? 

Authors' response:

Thank you for your comment. We replaced this with sex and/or gender (Page 19).

Reviewer's comment:

o Why is “boy” not included in Table 1 under “related terms mentioned”

Authors' response:

Thank you for noticing this. In fact, “Boy” was not used in any study. We added a sentence to this effect. (Page 18, line 407).

Reviewer comment:

• Language unclear/vague/incorrect in some sentences: i.e., Line 293-294 “Authors used sex as non-binary variable in no study”. Same for lines 307-308, 331-332, 373, etc.

Authors's response:

Thank you for your comment. We corrected this: “e.g. Authors did not use gender as a non-binary variable » line 441. The whole section has been revised by an English-language editor (Page 20, line 441).

Reviewer's comment:

• Table 2:

o What is “respect of the criteria (2 and 3)” representing? What does this mean? 

Authors's response: We removed this sentence.

Reviewer's comment:

• Line 354: “Countries” should be “regions” throughout. 

Author's response:

We replaced “countries” by “regions” throughout the text (Page 16, line 344 to page 29 (table 3)).

Reviewer's comment:

Discussion

• Overall, the discussion should be shortened/tightened, remove speculation, and provide MUCH more specific commentary. A thorough grammar review is also warranted, especially on page 24 

Authors's response:

Thank you for your comments, we completely rewrote our discussion as suggested and revised the language (Pages 30-38; lines 600-798).

Reviewer's comment:

• More is needed to discuss the implications of sex and gender (measurement and reporting) for shared decision-making. Why does sex/gender matter for SDM? What has this review taught us about state of sex/gender measurement/reporting in SDM intervention studies? What are the implications of these findings for future SDM work? The authors mention these things but very vaguely – instead, the authors should add detail to provide information on specific findings from studies that have established the importance of sex/gender for SDM, and specific suggestions for how future SDM research should measure/report on sex/gender as a result of these findings, and why 

Authors's response:

Thank you for your comments, which have helped us completely revise the discussion (see below) (Pages 30-38; lines 600-798).

Reviewer's comment:

• Feels that discussion could be broadly restructured/focused as follows:

o First: succinct narrative summary of main findings: no study met all three criteria for correct reporting on sex/gender. This did not vary by measured study characteristics.

o Second: implications of this finding for existing SDM research, and future SDM research

o Third: strengths/limitations

o Fourth: conclusions � recommendations from authors based on findings

• More specifically: authors could recommend measures of sex and gender that SDM researchers could use going forward – there is a rich literature on measurement of sex and gender identity that they could draw from and cite.

Authors' response:

Thank you for your comment. We restructured the discussion as follows:

First paragraph – narrative summary of findings.

Second paragraph, implications of findings, especially the interchangeable use of sex and gender, and the fact that time and place of publication etc. seem to make no difference.

Third, implications of the fact that no studies used non-binary variables: Non-binary people are an important and increasingly vocal segment of the population with their own specific physical and mental health issues. Many already feel excluded by the health system, and being invisible in research results will be a further alienating factor »

Third, a paragraph that deals with the measuring of sex (especially non-binary) and gender and discusses whether gender is better captured using categorical or continuous data, with examples. It also mentions the extra difficulty of measuring gender in secondary data analyses and proposes solutions.

Finally, a paragraph proposing how to capture sex and gender data (often interrelated) in questionnaires (Pages 30-38; lines 600-798).

---

## [Editor Report · Decision Letter 1]

25 Sep 2020

Sex and gender considerations in implementation interventions to promote shared decision making: A secondary analysis of a cochrane systematic review

PONE-D-20-16595R1

Dear Dr. Légaré,

We’re pleased to inform you that your manuscript has been judged scientifically suitable for publication and will be formally accepted for publication once it meets all outstanding technical requirements. 

Thank you for revising this work and making it even clearer and more comprehensible.

Kind regards,

Robert S. Phillips

Academic Editor

PLOS ONE
---

## [Editor Report · Acceptance letter]

29 Sep 2020

PONE-D-20-16595R1 

Sex and gender considerations in implementation interventions to promote shared decision making: A secondary analysis of a Cochrane systematic review 

Dear Dr. Légaré:

I'm pleased to inform you that your manuscript has been deemed suitable for publication in PLOS ONE. Congratulations! Your manuscript is now with our production department. 

Kind regards, 

on behalf of

Dr Robert S. Phillips 

Academic Editor

PLOS ONE